# Discovery of isoflavone phytoalexins in wheat reveals an alternative route to isoflavonoid biosynthesis

Guy Polturak [1,2,8] ✉, Rajesh Chandra Misra [1,8], Amr El-Demerdash [1,3], Charlotte Owen [1], Andrew Steed[4], Hannah P. McDonald[5], JiaoJiao Wang[1,6], Gerhard Saalbach[7], Carlo Martins [7], Laetitia Chartrain[4], Barrie Wilkinson [5], Paul Nicholson [4] & Anne Osbourn [1] ✉

Isoflavones are a group of phenolic compounds mostly restricted to plants of the legume family, where they mediate important interactions with plant-associated microbes, including in defense from pathogens and in nodulation. Their well-studied health promoting attributes have made them a prime target for metabolic engineering, both for bioproduction of isoflavones as high-value molecules, and in biofortification of food crops. A key gene in their biosynthesis, isoflavone synthase, was identified in legumes over two decades ago, but little is known about formation of isoflavones outside of this family. Here we identify a specialized wheat-specific isoflavone synthase, TaCYP71F53, which catalyzes a different reaction from the leguminous isoflavone synthases, thus revealing an alternative path to isoflavonoid biosynthesis and providing a non-transgenic route for engineering isoflavone production in wheat. *TaCYP71F53* forms part of a biosynthetic gene cluster that produces a naringenin-derived *O*-methylated isoflavone, 5-hydroxy-2′,4′,7-trimethoxyisoflavone, triticein. Pathogen-induced production and in vitro antimicrobial activity of triticein suggest a defense-related role for this molecule in wheat. Genomic and metabolic analyses of wheat ancestral grasses further show that the triticein gene cluster was introduced into domesticated emmer wheat through natural hybridization ~9000 years ago, and encodes a pathogen-responsive metabolic pathway that is conserved in modern bread wheat varieties.

Flavonoids are a large family of phenolic compounds that are ubiquitous in plants, where they have multiple functions, including in regulation of development, UV protection, and plant-microbe interactions[1]. A major role for flavonoids is in providing defense against pathogens, either through activity of preformed molecules (phytoanticipins) or via de novo synthesis of molecules following pathogen infection or wounding (phytoalexins)[2]. Isoflavones form part of the isoflavonoid subclass of flavonoids which, like other groups of flavonoids, have also been implicated in plant defense against pathogens[2].

[1]Biochemistry and Metabolism Department, John Innes Centre, Norwich NR4 7UH, UK. [2]Institute of Plant Sciences and Genetics in Agriculture, The Hebrew University of Jerusalem, Rehovot 7610001, Israel. [3]Division of Organic Chemistry, Department of Chemistry, School of Sciences, Mansoura University, Mansoura 35516, Egypt. [4]Crop Genetics Department, John Innes Centre, Norwich NR4 7UH, UK. [5]Molecular Microbiology Department, John Innes Centre, Norwich NR4 7UH, UK. [6]Tsinghua-Peking Joint Center for Life Sciences, and School of Life Sciences, Tsinghua University, Beijing 100084, China. [7]Proteomics Facility, John Innes Centre, Norwich NR4 7UH, UK. [8]These authors contributed equally: Guy Polturak, Rajesh Chandra Misra. ✉e-mail: guy.polturak@mail.huji.ac.il; anne.osbourn@jic.ac.uk

Isoflavones are most commonly found in the legume (Fabaceae) family, of which soybean is the main source in the human diet, although various isoflavone compounds have also been detected in numerous non-leguminous species[3,4], including wheat, in which low levels of genistein and daidzein[5,6], or of formononetin derivatives[7,8], were previously reported. Genes taking part in the biosynthesis of these or other isoflavones in wheat have not been identified to date, to the best of our knowledge. Previous studies of wheat flavonoid metabolism, did however lead to identification of genes/enzymes involved in biosynthesis of other groups of flavonoids, such as those catalyzing O-methylation reactions that form the flavone tricin[9] and its downstream glycosylation and acylation[10]. Recent mGWAS/mQTL studies in wheat also led to identification of various additional flavonoid-related candidate genes[10–12]. The occurrence of other flavonoids and flavonoid-related biosynthetic genes in wheat was recently reviewed[13].

Isoflavones are widely regarded to have health-promoting properties[14]. The 100-fold lower levels of isoflavonoids in diets typically present in 'Western' countries compared to those in Asia (the latter being rich in soy protein products) has been inversely correlated with breast cancer by epidemiological studies, and the beneficial effects of isoflavonoids in the prevention of cardiovascular disease, breast and prostate cancer, and postmenopausal symptoms have been further substantiated in preclinical studies[15]. The health-promoting, phytoestrogenic properties of isoflavones have long led to interest in these compounds as targets for metabolic engineering[16,17], and discovery of isoflavone synthase (IFS) from legumes[18–20] enabled engineered production of isoflavones in other non-producing plant species via heterologous IFS expression[21]. However, little is known about formation of isoflavones outside of the Fabaceae and specifically with regards to the key isoflavone synthase step, in which the isoflavonoids branch out from other flavonoid groups.

Here, functional characterization of a pathogen-induced biosynthetic gene cluster (BGC) in bread wheat reveals a metabolic pathway that produces an O-methylated isoflavone, 5-hydroxy-2′,4′,7-trimethoxyisoflavone, hereinafter named triticein. The five-step pathway to triticein from its precursor naringenin is fully elucidated through heterologous gene expression, in vitro enzymatic assays, and metabolic analyses of mutant wheat lines. The isoflavone synthase activity leading to formation of triticein is catalyzed by a cytochrome P450 enzyme, TaCYP71F53, in a different reaction from the currently known isoflavone synthase from legumes, thus uncovering an alternative route to biosynthesis of isoflavonoids. The presence of an active triticein BGC in the wild grass Aegilops tauschii indicates that triticein biosynthesis was incorporated into modern bread wheat through the hybridization between A. tauschii and the ancestral tetraploid emmer wheat, thus illustrating how metabolic enrichment of specialized metabolism in plants can be facilitated through hybridization-derived polyploidy. Identification of the wheat isoflavone synthase provides a route for fortification of wheat with health-promoting isoflavonoids, without the use of transgenics.

## Results

### A pathogen-induced biosynthetic gene cluster in wheat encodes a putative flavonoid metabolic pathway

We previously reported the identification of six pathogen-induced biosynthetic gene clusters in Triticum aestivum (hexaploid bread wheat), of which one cluster, named BGC4(5D) (Fig. 1a), was predicted to encode a flavonoid biosynthetic pathway[22]. Transient expression experiments in Nicotiana benthamiana implicated the involvement of at least six genes from the cluster (TaCHS1, TaOMT3, TaOMT8, TaCYP71C164, TaCYP71F53, and TaOMT6,) in production of an unidentified O-methylated flavonoid ($m/z = 329.1012$, $[M + H]^+$, $C_{18}H_{17}O_6$, calc. 329.1019, $\Delta = -2.33$ ppm) the pathway's putative end-product. An additional lowly-expressed chalcone synthase and a gene annotated as

a chalcone isomerase (chi-D1) were found in the cluster, for which involvement in the same pathway could not be clearly verified[22]. The BGC4(5D) locus also includes three genes with functional annotations unlikely to be associated with flavonoid metabolism and that are not co-expressed with the remaining genes (Supplementary Table S1), which were thus excluded from our subsequent functional analyses.

To determine whether the heterologously produced O-methylated flavonoid can also be detected in wheat, extracts from wheat blades (cv. Chinese Spring) were analyzed by liquid chromatography-mass spectrometry (LC-MS), allowing detection of the corresponding $m/z = 329.1012$ mass signal in the wheat extracts. Furthermore, blades treated with the defense-response elicitor methyl jasmonate (MeJA) resulted in >20-fold higher accumulation of this flavonoid, compared to non-treated leaves (Fig. 1b). The retention time and fragmentation patterns of the $m/z = 329.1012$ compound detected in the wheat extracts corresponded with those of the N. benthamiana-produced compound (Supplementary Fig. S1), thus confirming the occurrence and inducible production of the investigated flavonoid in wheat. Increased accumulation of the compound could also be observed in pathogen-infected wheat plants; stem bases of 3-week-old plants were inoculated with Fusarium culmorum (Supplementary Fig. S2), a fungal pathogen causing foot rot, root rot and head blight in various cereals, including wheat[23]. Blade extracts from F. culmorum-infected plants accumulated ~30-fold more of the flavonoid compound compared to mock-treated plants, 7- or 14-days post infection (Fig. 1c). Correspondingly, quantitative real-time PCR (qRT-PCR) analysis of the mock and F. culmorum-infected plants showed significant induction of all analyzed cluster genes 7 days post infection, with the exception of chi-D1 (Fig. 1d). The F. culmorum- and MeJA- induced production was consistent with a role for this flavonoid as a phytoalexin (i.e., a pathogen-induced small molecule) in wheat. This was further supported by analysis of gene expression in several publicly available RNA-seq datasets from biotic stress experiments, which include infection with fungal or bacterial pathogens, or treatment with pathogen associated molecular patterns (PAMPs). The data generally show induction of TaCHS1, the two cytochrome P450 genes TaCYP71C164 and TaCYP71F53, and the four O-methyltransferases TaOMT3, TaOMT6, TaOMT7, and TaOMT8 under the different treatments (Supplementary Fig. S3). As observed with the F. culmorum infection, chi-D1 generally shows only weak or no induction by biotic stress in these data (Supplementary Fig. S4). Examination of BGC4(5D) gene expression in an extensive developmental/time course wheat transcriptome dataset[24] further showed a noticeably different expression pattern for chi-D1 compared to other genes in the cluster, that are generally co-expressed (Supplementary Fig. S5). To investigate the possible involvement of CHI genes that are not part of BGC4(5D) in this biosynthetic pathway, we identified additional putative chalcone flavanone-isomerase candidates by a Pfam domain search of the wheat proteome, and examined the expression patterns of their coding genes within a biotic stress-related gene expression dataset which we have previously reported[22]. None of the 10 putative CHI genes that we identified were found to be pathogen-induced. Similarly, inspection of the expression patterns of 25 putative CHS genes showed that only TaCHS1 is upregulated following pathogen infection (Supplementary Fig. S5).

### BGC4(5D) encodes a sequential biosynthetic pathway for an O-methylated isoflavone

To functionally analyze the BGC4(5D) cluster, we first set out to identify the $m/z = 329.1012$ flavonoid by purifying the compound from agroinfiltrated N. benthamiana leaves and determining its structure by extensive 1D and 2D NMR analyses. To this end, a mixture of Agrobacterium tumefaciens strains harboring single-gene expression plasmids for seven genes from the BGC4(5D) cluster was vacuum-infiltrated into the leaves of ~100 N. benthamiana plants. The seven

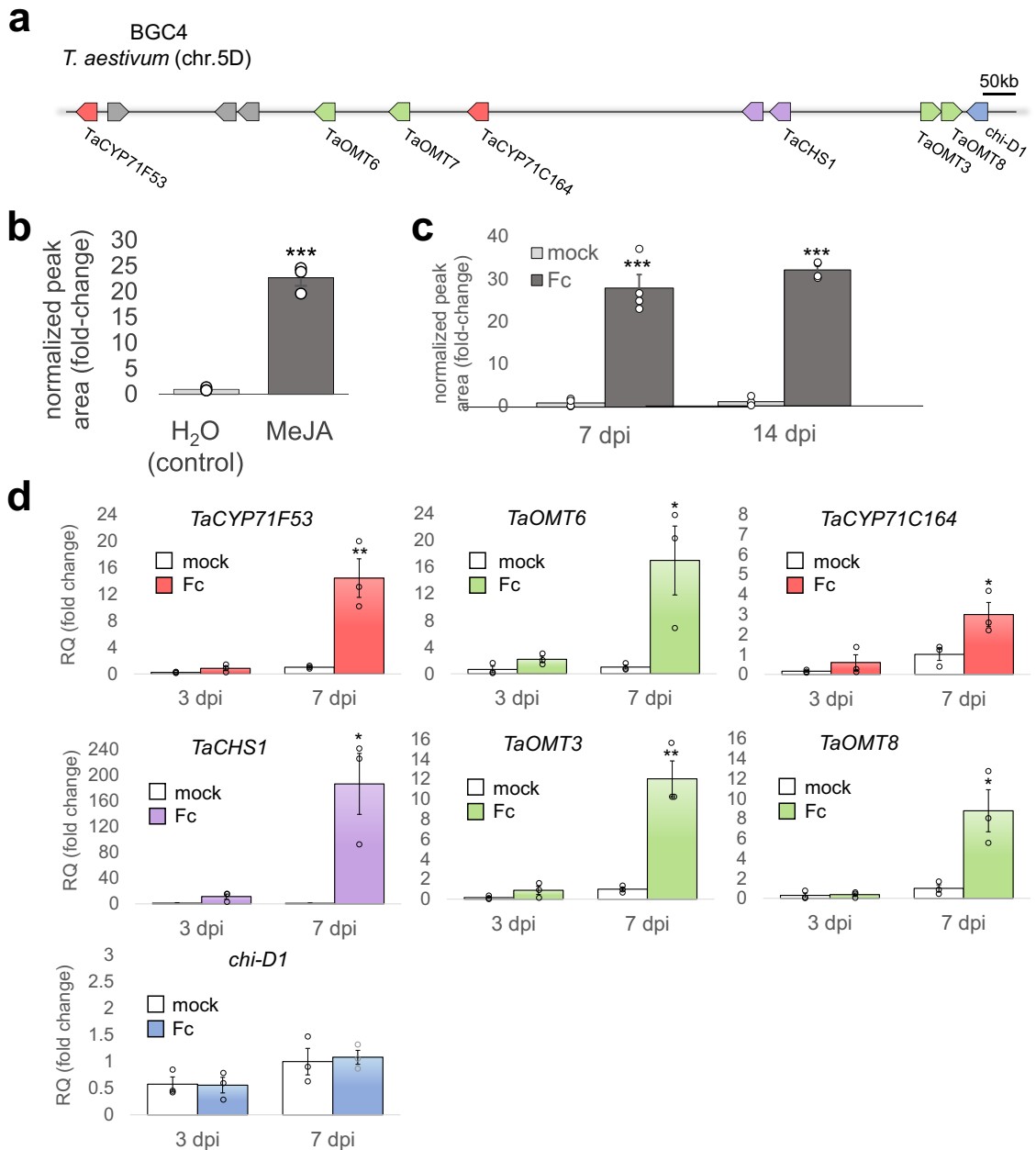

**Fig. 1 | BGC4(5D) is induced by methyl jasmonate and by fungal pathogen *Fusarium culmorum*. a** illustration of the gene cluster on wheat chromosome 5D. Red, cytochrome P450s; green, *O*-methyltransferases; purple, chalcone synthases; blue, chalcone isomerase; gray, other intervening genes. Kb: kilobase. **b** Relative quantification of ($m/z$ = 329.1012) ion in extracts from wheat blades treated with methyl jasmonate (MeJA) or water (control). $p$-val = $1.58 \times 10^{-4}$. **c** Relative quantification of ($m/z$ = 329.1012) ion in blade extracts from mock- or *F. culmorum*-treated wheat plants, 7- and 14-days post infection. Fc, *Fusarium culmorum*.

$p$-val = $1.47 \times 10^{-4}$ (7 dpi), $1.21 \times 10^{-7}$ (14 dpi). **d** Quantitative real-time PCR of seven genes from cluster BGC4(5D) in blade extracts from mock- or *F. culmorum*- treated wheat plants, 3- and 7-days post infection. Fc, *Fusarium culmorum*. 7 dpi $p$-val = 0.0099 (*TaCYP71F53*), 0.0365 (*TaOMT6*), 0.0420 (*TaCYP71C164*), 0.0171 (*TaCHS1*), 0.0036 (*TaOMT3*), 0.0221 (*TaOMT8*), 0.7893 (*chi-D1*). For panels **b**, **c**, and **d**, relative quantification (fold-change) values indicate means of three biological replicates ± SEM. Asterisks denote the statistical significance of a two-tailed *t*-test. *$P < 0.05$, **$P < 0.01$, ***$P < 0.001$. Source data are provided as a Source Data file.

genes are annotated as a chalcone synthase (*TaCHS1*), the non-induced chalcone isomerase (*chi-D1*), two cytochrome P450s (*TaCYP71C164*, *TaCYP71F53*) and three *O*-methyltransferases (*TaOMT3*, *TaOMT6*, *TaOMT8*). The cluster genes were expressed together with the *Arabidopsis thaliana MYB12* transcription factor, which was previously shown to significantly enhance production of flavonoids and other phenylpropanoids when expressed in tobacco and tomato[25], through upregulation of key genes in the phenylpropanoid pathway and its upstream pathways[26]. Preliminary analysis confirmed that expressing CaMV 35S::*AtMYB12* together with the BGC4(5D) cluster genes indeed

leads to a 6.5-fold increase in yield of the target compound, as well as a >2-fold increased accumulation of naringenin, a predicted precursor of the target molecule (Supplementary Fig. S6). Large-scale infiltration of *A. tumefaciens* strains harboring expression constructs for the same set of seven genes in leaves of ~100 *N. benthamiana* plants, followed by metabolite extraction and compound purification from 86 g of dried leaf biomass, yielded 12.5 mg of white powder. Surprisingly, integrated NMR analyses and subsequent comparison with NMR spectra from the literature revealed that the purified compound is an isoflavone rather than an expected flavone. Specifically, the compound was identified by

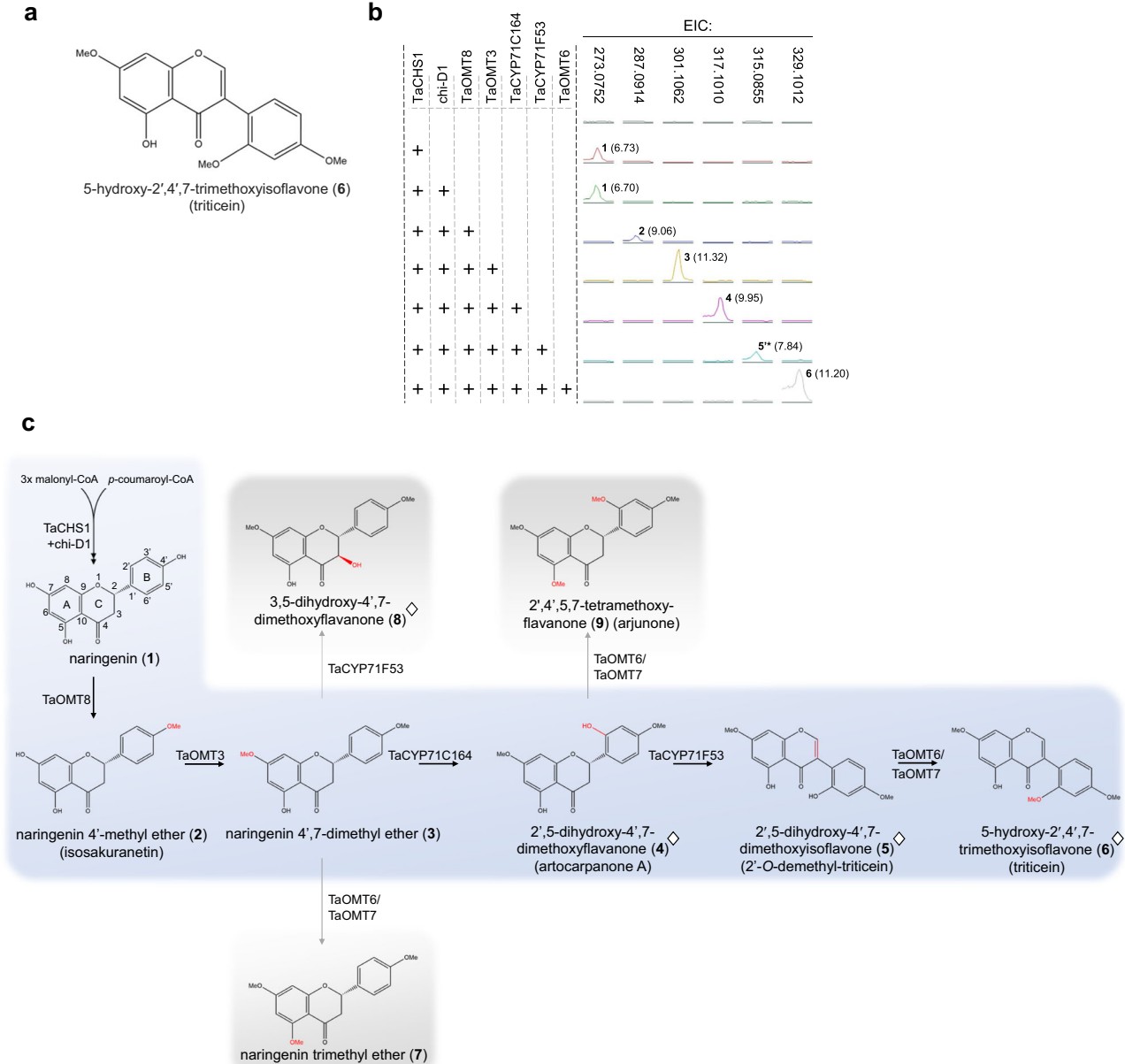

**Fig. 2 | Wheat gene cluster BGC4(5D) encodes an isoflavone biosynthetic pathway. a** NMR-assigned chemical structure of 5-hydroxy-2′,4′,7-trimethoxyiso-flavone (triticein) (**6**), the putative end product of the BGC4(5D)-encoded pathway. **b** Extracted LC-MS chromatograms of leaf extracts from *Nicotiana benthamiana* combinatorial transient gene expression experiments. EIC: extracted ion chromatogram. Peak numbering corresponds with compound structures presented in panel (**c**) and Supplementary Table S2. Peak retention times are shown in brackets.

Plus sign indicates presence of gene in the expressed gene combination. **c** Proposed pathway for triticein biosynthesis from naringenin. Modifications in each step are highlighted in red. Naringenin structure shows standard flavonoid carbon numbering and ring nomenclature. Structures marked with a diamond sign were assigned by NMR analyses. Side products of the pathway are highlighted in grey boxes.

extensive 1D and 2D NMR as 5-hydroxy-2′,4′,7-trimethoxyisoflavone (**6**) (Fig. 2a, Supplementary Data 1), here named triticein. This compound is not currently known to occur in wheat or other grasses, but was previously reported in the leguminous shrub *Dalbergia vacciniifolia*, native to tropical East Africa[27].

Following the structural assignment of the isoflavone product, we carried out further combinatorial transient expression experiments in *N. benthamiana* in order to characterize the functions of the enzymes encoded by the triticein-forming BGC and delineate the resulting metabolic pathway. The seven cloned genes were thus transiently expressed in different combinations in *N. benthamiana* leaves, and leaf extracts were subjected to LC-MS analyses. Combined expression of *TaCHS1* and *chi-D1* resulted in production of naringenin (**1**)

($m/z = 273.0752$, [M + H]⁺, $C_{15}H_{13}O_5$, calc. 273.0757, Δ=−2.14 ppm), confirmed by comparison to a commercial standard (Supplementary Fig. S7). Interestingly, *chi-D1* expression was not essential for accumulation of naringenin (Fig. 2b), presumably due to activity of an endogenous chalcone isomerase in *N. benthamiana*, or spontaneous isomerization of naringenin chalcone[28], but was nevertheless included in other combinatorial experiments described below. Naringenin then undergoes two consecutive *O*-methylations, firstly 4′-*O*-methylation by TaOMT8, to form naringenin 4′-methyl ether (**2**) (isosakuranetin) ($m/z = 287.0914$, [M + H]⁺, $C_{16}H_{15}O_5$, calc. 287.0914, Δ = −0.11 ppm), then 7-*O*-methylation by TaOMT3, to form naringenin 4′,7-dimethyl ether (**3**) ($m/z = 301.1062$, [M + H]⁺, $C_{17}H_{17}O_5$, calc. 301.1070, Δ = −2.72 ppm) (Fig. 2b, c). Both compounds were identified by comparison to

commercial standards (Supplementary Fig. S7). Notably, there seems to be no branching in this upstream part of the pathway; only the addition of *TaOMT8* to the expressed *TaCHS1* and *chi-D1* pair led to a substantially diminished naringenin peak and formation of a prominent new peak. Similarly, only further addition of *TaOMT3* to the gene combination effectively metabolized the TaOMT8 product- isosakuranetin (Supplementary Fig. S8).

The addition of *TaCYP71C164* to the gene combination resulted in formation of a new peak (**4**) ($m/z = 317.1010$, $[M + H]^+$, $C_{17}H_{17}O_6$, calc. 317.1019, $\Delta = -3.10$ ppm) +16 amu (atomic mass units) from the observed mass for naringenin 4′,7-dimethyl ether ($m/z = 301.1062$), suggesting a probable hydroxylation catalyzed by this cytochrome P450 (Supplementary Fig. S9). To identify the product of the *TaCYP71C164* modification, we again carried out large-scale infiltration of combined *A. tumefaciens* strains carrying expression constructs of the five genes (*TaCHS1, chi-D1, TaOMT8, TaOMT3, TaCYP71C164*) together with *AtMYB12*, in ~60 *N. benthamiana* plants. Extraction and purification from 101 g dry leaf material yielded 2.2 mg of the purified compound. NMR analysis of the resulting product revealed it to be 2′,5-dihydroxy-4′,7-dimethoxyflavanone (artocarpanone A) (**4**) (Supplementary Data 1). TaCYP71C164 was thus identified as a flavanone 2′-hydroxylase. Similarly to artocarpanone A, triticein also has an oxygen incorporated at the 2′ position, consistent with artocarpanone A being a pathway intermediate. Since the product of TaCYP71C164 was a flavanone rather than an isoflavanone, it was assumed that TaCYP71F53 would catalyze both the aryl migration and C-ring desaturation required to ultimately obtain the isoflavone triticein. Addition of *TaCYP71F53* to the expressed gene combination indeed resulted in formation of a new peak (**5**) ($m/z = 315.0855$, $[M + H]^+$, $C_{17}H_{15}O_6$, calc. 315.0863, $\Delta = -2.55$ ppm), −2 amu from artocarpanone A (Supplementary Fig. S9), corresponding with a flavanone to flavone conversion (i.e. C-2/C-3 desaturation of the C-ring). Also notable was a distinct shift in UV absorbance of the TaCYP71F53 product, compared to its upstream pathway intermediates; while naringenin (**1**), isosakuranetin (**2**) and naringenin 4′,7-dimethyl ether (**3**) all exhibited a $\lambda_{max}$ of 288 nm, the TaCYP71F53 product absorbs at $\lambda_{max} = 260$ nm, a property shared with triticein (**6**) (Supplementary Fig. S10). Full NMR analysis of the compound, purified from ~100 vacuum-infiltrated leaves of *N. benthamiana*, showed the structure to be triticein lacking 2′-O methylation, namely 2′,5-dihydroxy-4′,7-dimethoxyisoflavone (**5**), hereinafter named 2′-O-demethyl-triticein (Supplementary Data 1), thereby confirming that *TaCYP71F53* codes an isoflavone synthase. In the final reaction of the proposed pathway, TaOMT6 methylates the 2′-OH group of 2′-O-demethyl-triticein (**5**) to form 5-hydroxy-2′,4′,7-trimethoxyisoflavone, triticein (**6**) (Fig. 2b, c, Supplementary Fig. S9). A fourth O-methyltransferase in the gene cluster, *TaOMT7*, is a tandem duplicate of *TaOMT6* with only one amino acid difference in its protein sequence (G72R) and was therefore not included in the initial combinatorial expression experiments. Subsequent transient expression experiments showed that similarly to TaOMT6, TaOMT7 catalyzes the conversion of 2′-O-demethyl-triticein (**5**) to triticein (**6**) with comparable efficiency (Supplementary Fig. S11). A maximum-likelihood phylogenetic tree of putative protein sequences of 165 wheat O-methyltransferases shows that *TaOMT6* and *TaOMT7* are distant from the two additional OMTs in BGC4(5D), *TaOMT3* and *TaOMT8*, which reside in separate but closely related groups (Supplementary Fig. S12).

## Modularity of the BGC4(5D)-encoded pathway yields side-products in *N. benthamiana*

The combinatorial transient expression experiments in *N. benthamiana* revealed a linear biosynthetic pathway from naringenin (**1**) to triticein (**6**). However, pathway branching towards formation of side-products was also observed, specifically with methylation of two additional substrates by TaOMT6; exclusion of *TaCYP71F53* from the combination of expressed genes resulted in formation of a new peak,

putatively identified as 2′,4′,5-7-tetramethoxyflavanone (arjunone) (**9**) ($m/z = 345.1324$, $[M + H]^+$, $C_{19}H_{21}O_6$, calc. 345.1332, $\Delta = -2.46$ ppm), the product of 2′-O and 5-O methylation of artocarpanone A (**4**) (Fig. 2c, Supplementary Fig. S9). Exclusion of both CYP71s from the combination leads to formation of a peak identified as naringenin trimethyl ether (**7**) ($m/z = 315.1221$, $[M + H]^+$, $C_{18}H_{19}O_5$, calc. 315.1227, $\Delta = -2.00$ ppm) based on comparison to a commercial standard- the resulting product of 5-O methylation of naringenin 4′,7-dimethyl ether (**3**) (Fig. 2c, Supplementary Fig. S7, Supplementary Fig. S9).

Intriguingly, TaCYP71F53 shows substantially different catalytic activity when *TaCYP71C164* and *TaOMT6* are excluded from the gene expression combination, resulting in formation of a new peak (**8**) (Supplementary Fig. S9). Large-scale infiltration of ~60 *N. benthamiana* plants with this gene combination, followed by extraction and purification from 103 g dry leaf material yielded 6.6 mg of the product, identified as 3,5-dihydroxy-4′,7-dimethoxyflavanone (**8**) ($m/z = 317.1012$, $[M + H]^+$, $C_{17}H_{17}O_6$, calc. 317.1019, $\Delta = -2.33$ ppm) by 1D and 2D NMR analyses (Fig. 2c, Supplementary Fig. S7, Supplementary Data 1). TaCYP71F53 thus appears to depend on the presence of the 2′-hydroxy group in its substrate for catalyzing isoflavone synthase activity. A tabular summary of peaks representing compounds **1–9** identified in the transient expression experiments described above is provided in Supplementary Table S2. NMR spectra for compounds (**4**), (**5**), (**6**), (**8**) are provided in Supplementary Data 1.

## In vitro enzymatic assays of the two CYP71s from BGC4(5D)

The enzymatic activities of TaCYP71C164 and TaCYP71F53 were further investigated by expression in yeast (*Saccharomyces cerevisiae*) and in vitro enzymatic assays. WAT11 yeast strains expressing each of the CYP71 genes, both CYP71s, or neither, were generated. Microsomal fractions from these yeast strains were used in assays containing either naringenin 4′,7-dimethyl ether (**3**), or artocarpanone A (**4**) as substrates. In assays with naringenin 4′,7-dimethyl ether (**3**), only microsomal fractions containing TaCYP71C164 were found to produce artocarpanone A (**4**) (Fig. 3a, Supplementary Fig. S13). In parallel, in reactions with the substrate artocarpanone A (**4**), only the two fractions containing TaCYP71F53 were found to produce 2′-O-demethyl-triticein (**5**) (Fig. 3b). LC-MS chromatograms of the TaCYP71C164 and control strain (empty vector) reactions containing artocarpanone A also showed a peak with a mass and retention time similar to 2′-O-demethyl-triticein (**5**), which was determined to be a different molecule, based on the different MS2 fragmentation patterns of the two compounds (Supplementary Fig. S13). These enzymatic assays thus further validated the respective flavanone 2′-hydroxylase and isoflavone synthase enzymatic activities of TaCYP71C164 and TaCYP71F53, deduced from the *N. benthamiana* transient expression experiments.

## *TaCYP71F53, TaOMT3,* and *TaOMT8* mutant lines lack triticein and accumulate their respective predicted substrates

Further support for the proposed activity of TaCYP71F53 was obtained by metabolic analysis of mutant wheat lines sourced from a TILLING population of hexaploid wheat (in a cv. Cadenza background)[29,30]. Two independent TILLING lines Cad0227 and Cad1793, were identified as carrying mutations that result in a premature stop codon in the *TaCYP71F53* coding sequence (Supplementary Fig. S14). LC-MS analysis of blade extracts from KASP-genotyped, MeJA-treated plants showed that triticein (**6**) and 2′-O-demethyl-triticein (**5**) are found in parent 'Cadenza' plants but cannot be detected in plants from the Cad0227 and Cad1793 mutant lines (Fig. 3c). Corresponding with the absence of these compounds was an observed accumulation of the proposed TaCYP71F53 substrate, artocarpanone A (**4**), together with an apparent reduction in levels of naringenin 4′,7-dimethyl ether (**3**) vs. the parental Cadenza line. Mutant lines harboring premature stop codons for *TaOMT8* (Cad1684) and *TaOMT3* (Cad1682) were also examined and,

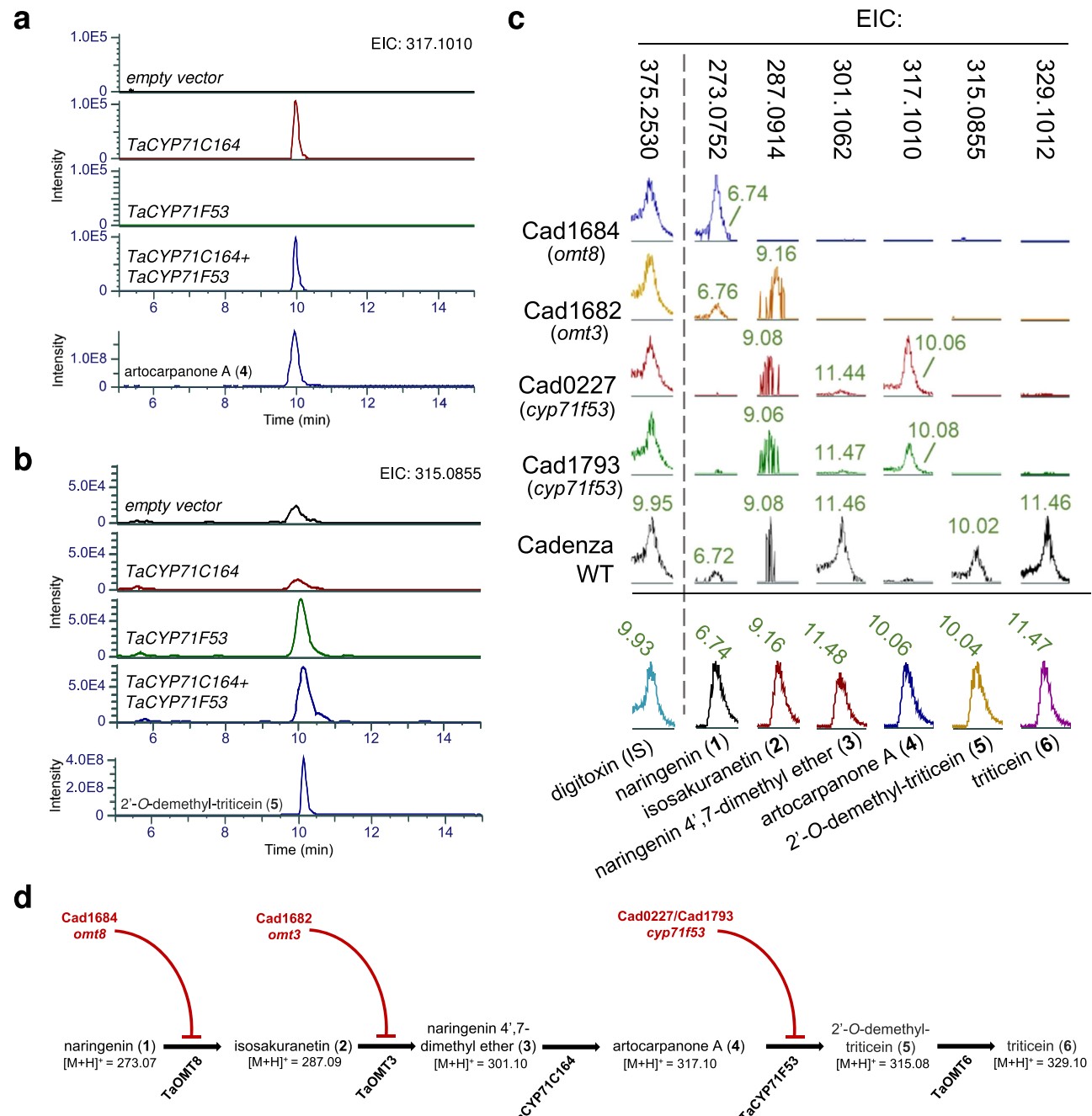

**Fig. 3 | Functional analysis of BGC4(5D) genes by in vitro enzymatic assays and metabolic analysis of wheat mutant lines. a** LC-MS chromatograms of in vitro enzymatic reactions with naringenin 4',7-dimethyl ether (**3**) as substrate, and microsomal fractions from yeast expressing *TaCYP71C164* and/or *TaCYP71F53*. Extracted ion chromatogram (EIC) for molecular [M + H]⁺ ion of the predicted reaction product artocarpanone A (**4**) (*m/z* = 317.1010) is shown. *Y*-axes are linked. Chromatogram of artocarpanone A (**4**) purified compound is given for reference. **b** Chromatograms of in vitro enzymatic reactions with artocarpanone A (**4**) as substrate, and microsomal fractions from yeast expressing *TaCYP71C164* and/or *TaCYP71F53*. Extracted ion chromatogram (EIC) for the [M + H]⁺ ion of the predicted reaction product 2'-*O*-demethyl-triticein (**5**) (*m/z* = 315.0855) is shown. *Y*-axes are linked. Minor peaks observed in the empty vector and *TaCYP71C164* samples originate from a different compound with similar retention time and *m/z*

to compound (**5**) (Supplementary Fig. S13). Chromatogram of 2'-*O*-demethyl-triticein (**5**) purified compound is given for reference. **c** Extracted LC-MS chromatograms of blade extracts from MeJA-treated plants of TILLING mutants (Cad1684, *omt8*; Cad1682, *omt3*; Cad0227/Cad1793, *cyp71f53*) and 'Cadenza' wildtype parental line. Chromatograms for commercial standards (naringenin (**1**), isosakuranetin (**2**), naringenin 4',7-dimethyl ether (**3**) and purified compounds (artocarpanone A (**4**), 2'-*O*-demethyl-triticein (**5**), and triticein (**6**) are shown in bottom row, for reference. Digitoxin was used as an internal standard (IS) in all samples. Extracted ion chromatograms (EIC) are for ions representing the compounds specified above. *Y*-axes are linked for chromatograms of each extracted mass, but not between different masses. **d** Illustration of proposed triticein biosynthetic pathway, impeded by loss-of-function mutations of *TaOMT8*, *TaOMT3*, or *TaCYP71F53*.

similarly to the *TaCYP71F53* mutant lines, showed accumulation of the proposed respective substrates (naringenin (**1**), isosakuranetin (**2**)) and absence of TaOMT3/TaOMT8 products and downstream metabolites (Fig. 3c, Supplementary Fig. S15). All peaks were identified by

comparison to commercial standards or purified compounds (Fig. 3c, Supplementary Fig. S16). Metabolic analysis of the Cadenza lines also enabled detection of two of the branching pathway side-products. 3,5-dihydroxy-4',7-dimethoxyflavanone (**8**) was found in wildtype Cadenza

but not in any of the four mutant lines (Supplementary Fig. S17), coinciding with the proposed pathway (Fig. 2c) in which *TaOMT3*, *TaOMT8* and *TaCYP71F53* are all necessary for its biosynthesis. The peak putatively identified as 2′,4′,5,7-tetramethoxyflavanone (**9**) in *N. benthamiana* heterologous expression experiments could also be detected (in trace amounts) in the two *TaCYP71F53* mutant lines (Supplementary Fig. S18). Naringenin trimethyl ether (**7**) was expectedly not found in the *TaOMT8* and *TaOMT3* mutants, but was also not detected in the *TaCYP71F53* mutant lines, presumably due to naringenin 4′,7-dimethyl ether (**3**) being more efficiently metabolized by *TaCYP71C164* than by *TaOMT6/TaOMT7*. This is consistent with results from the *N. benthamiana* experiments, in which a prominent peak for naringenin 4′,7-dimethyl ether (**3**) still remained with the additional expression of *TaOMT6* (Supplementary Fig. S9).

To assess possible contribution of the mutational backgrounds of the four Cadenza mutant lines to the observed metabolic phenotypes, the full set of characterized SNPs in each line was examined. Specifically, we searched for potentially high impact variations leading to premature stop codons, splice variations, transcript ablation, frameshift, or loss of start/stop codons, within *O*-methyltransferases in Cad1682 and Cad1684, and within cytochrome P450s in the Cad0227 and Cad1793 lines. Potentially deleterious variants were found in three and six CYPs in Cad0227 and Cad1793, respectively, of which only *TaCYP71F53* (*TraesCS5D02G487900*) was common to both lines. No potentially deleterious variations within additional OMTs were found in Cad1684 and Cad1682, other than a plausibly low impact mutation in *TaOMT6* in Cad1682, conferring a premature stop codon that would lead to a loss of 13 amino acids at the C-terminus of the protein (Supplementary Data 2).

Metabolic analysis of the mutant wheat lines thus supports the proposed enzymatic activities of TaCYP71F53, TaOMT3, and TaOMT8 (Fig. 3d). The absence of downstream pathway products in the mutant plants also indicates that the triticein biosynthetic pathway is encoded solely by BGC4(5D), and not by other homologs of the three mutated genes, whether clustered or dispersed across the genome. Also notable was the substantially higher abundance of triticein (**6**) compared to other pathway metabolites, as inferred by the 1 to 3 orders of magnitude higher signal intensity of the triticein $[M + H]^+$ ion. A similar pattern of triticein accumulation could also be observed in blade extracts of MeJA-treated 'Chinese Spring' plants (Supplementary Fig. S19), supporting the assignment of triticein as the putative pathway end-product. The Cad1682, Cad1684 and Cad1793 mutant lines were also subjected to pathogen susceptibility assays, in which detached leaves from plants carrying either the wildtype or the mutated allele were infected with the necrotroph *F. culmorum*, the hemibiotrophic cereal pathogen *Magnaporthe oryzae* (blast), or the biotrophic pathogen *Blumeria graminis f. sp. tritici* (powdery mildew). However, no obvious difference in disease severity between mutant and control plants was found in any of these assays, with any of the three lines (Supplementary Figs. S20, S21, S22). Cad0227 was excluded from these assays due to the mutant allele being fixed in the accessible germplasm from the TILLING population, thus preventing the use of an appropriate line as control.

## Triticein inhibits fungal spore germination and bacterial growth in vitro

Quantitative disease resistance in plants is typically conditioned by multiple loci that have partial effects on resistance[31]. The lack of observed differences in disease severity in triticein-deficient leaves could thus be due to masking by other genetic factors affecting colonization and disease progression. We therefore proceeded to examine potential antifungal activity of triticein in vitro, using a spore germination assay for the fungal phytopathogen *Botrytis cinerea* (strain B05.10). *B. cinerea* conidiospores were suspended in PDB media, to which a triticein solution in acetone was added (to a

final concentration of 100 μM). After incubation for 3 h, spores exhibited a 60% reduction in germination rate in the presence of 1% acetone with triticein, compared to acetone only (*p*-val < 0.0001) (Fig. 4a, b). In addition, a 25% reduction in the average germ tube lengths was measured after 5 h in spores germinating in the presence of 1% acetone with triticein, compared to acetone alone (*p*-val < 0.0001) (Fig. 4a, c).

Inspection of publicly available wheat gene expression data shows that in addition to induction by fungal pathogens, the BGC4(5D) cluster is also induced by bacterial pathogen infection, as well as by flagellin, a bacterial PAMP (Supplementary Fig. S2). We thus examined potential antibacterial activity of triticein with growth inhibition assays in liquid media, in which bacteria were grown in 96-well plates for 12 h, with $OD_{600}$ measured every 30 min. Assays were carried out with three bacterial species- the Gram-negative *Escherichia coli* (DH5α), and the Gram-positive *Bacillus subtilis* (168) and *Staphylococcus aureus* (ATCC 6538 P, methicillin-sensitive; MSSA). An additional *E. coli* strain was tested, NR698, which has a deficient outer membrane and thus typically exhibits higher sensitivity to antimicrobials compared to standard *E. coli* lab strains[32,33]. Growth inhibition was observed for all four strains, with the MSSA showing near complete inhibition in the presence of triticein in concentrations as low as 1 μM (0.33 μg·mL⁻¹). Growth of a methicillin-resistant *S. aureus* (MRSA) strain (BAA-1717) was also inhibited at a 1–25 μM range, but to a lower extent than the MSSA strain (Fig. 4d, Supplementary Fig. S23).

## Triticein biosynthesis is conserved in bread wheat ancestor *Aegilops tauschii*

We previously reported that gene clusters homologous to BGC4(5D) are not found in genomes of other grasses, including maize, oat, rice and *Brachypodium distachyon*[22], suggesting that occurrence of this gene cluster may be restricted to wheat and its close progenitors. Microsynteny analysis of the BGC4(5D) region shows that a homologous cluster is found in goat grass (*Aegilops tauschii*, DD genome) but not in wild emmer wheat (*Triticum turgidum ssp. dicoccoides*, AABB genome) consistent with the presence of the cluster solely in the D subgenome of hexaploid wheat (Fig. 5a). Examination of the homologous cluster on Chr. 5D of *A. tauschii* shows that it is highly conserved with wheat BGC4(5D) and contains 1-to-1 orthologs for each of the wheat genes implicated in triticein biosynthesis, with the exception of a single *A. tauschii* gene that corresponds with the functionally equivalent tandem duplicates *TaOMT6/TaOMT7* (Supplementary Fig. S24).

A search for orthologs of BGC4(5G) genes in additional Triticeae tribe genomes revealed the presence of a homologous cluster in additional 'D lineage' genomes of grasses from section Sitopsis, a group of five *Aegilops* species that contain the S genome and are also closely related to wheat. Notably, in addition to the lack of a full or partial homologous cluster in genomes outside the D lineage, single orthologs of the cluster genes could not be identified in any of the analyzed genomes, with the exception of *chi-D1*, which is conserved in sequence and position on chromosome 5 throughout the Triticeae genomes (Fig. 5b, Supplementary Data 3). Considering the different phylogenetic distribution of *chi-D1*, together with the different expression patterns exhibited by this gene compared to other BGC4(5D) genes (Fig. 1e, Supplementary Fig. S3, S4) and[22], this gene is likely not an integral part of the cluster. The *chi-D1* gene contains a predicted full protein sequence with conserved CHI motifs[34], and is a 1-to-1 ortholog of the functionally characterized rice chalcone isomerase *OsCHI* (*gh1*)[35], but its presumed enzymatic function as a naringenin chalcone isomerase remains to be verified.

The monophyletic distribution of the triticein cluster in D lineage species indicates that the cluster was formed after the divergence between the A and D lineages, which occurred ~6.75 million years ago (Fig. 5b)[36]. However, earlier formation in a more ancestral genome and

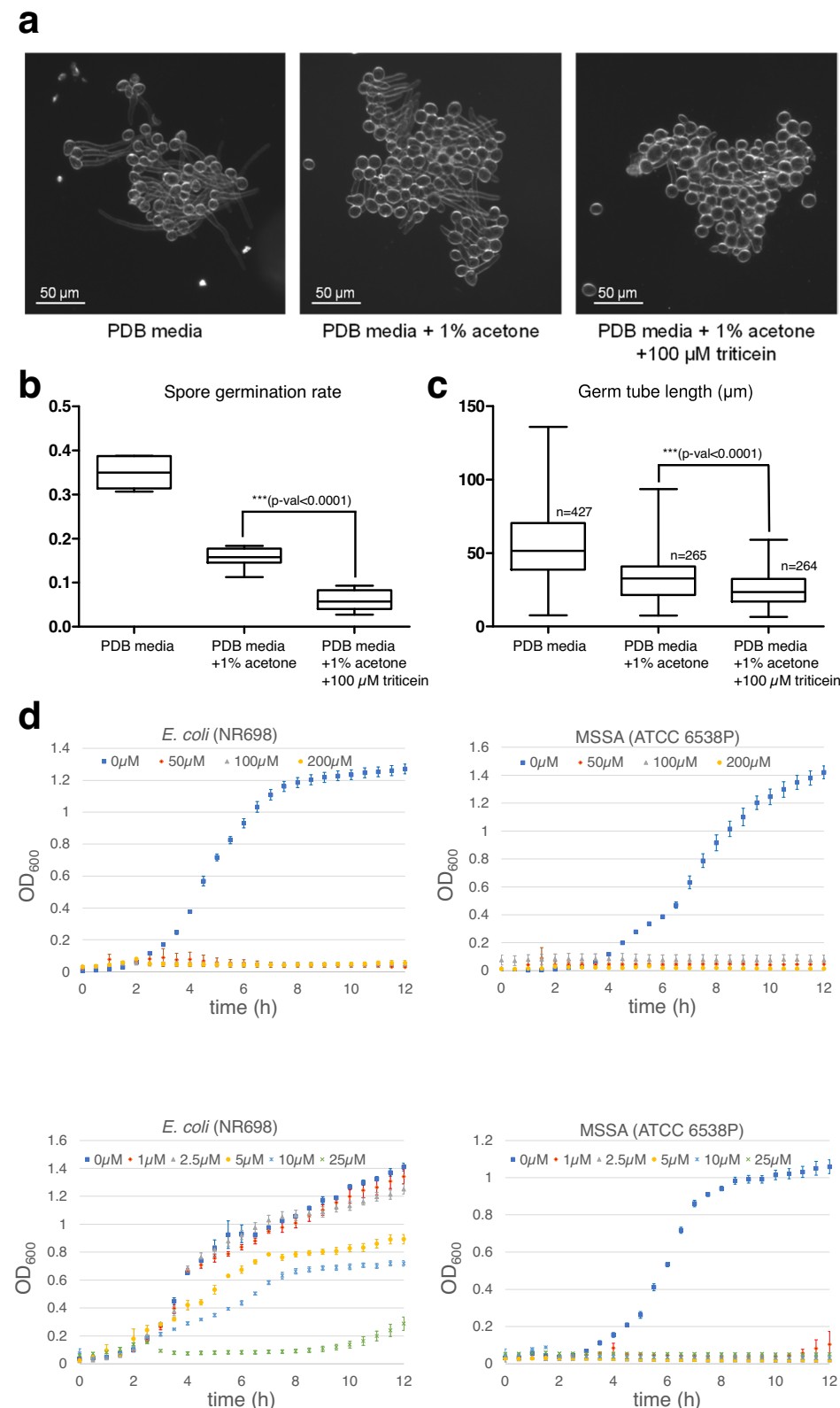

subsequent multiple losses of the cluster cannot be ruled out. The presence of a highly conserved cluster in *A. tauschii* implied that triticein biosynthesis may also occur in this plant. Analysis of MeJA-treated *A. tauschii* leaves indeed allowed detection of a compound identified as triticein (**6**) based on retention time, accurate mass and fragmentation pattern (Fig. 5c). In contrast, the absence of a gene cluster homologous to BGC4(5D) in the wild emmer genome suggests the lack of triticein production in emmer wheat. This lack of triticein

was confirmed by analysis of six domesticated and wild *T. turgidum* accessions, in none of which triticein could be detected in leaves of MeJA-treated plants (Fig. 5c).

## Discussion

Hexaploid bread wheat was formed only 8500–9000 years ago, through hybridization between the tetraploid domesticated emmer wheat (*Triticum turgidum ssp. dicoccum*) and *Aegilops tauschii*[37].

**Fig. 4 | Triticein inhibits bacterial growth and fungal spore germination.**
**a** Microscopy images of *Botrytis cinerea* (B05.10) conidiospores grown in PDB media, 5 h after addition of acetone containing 100 μM triticein (final concentration), acetone only, or no addition of solvent. **b** Distribution of *B. cinerea* spore germination rate, 3 h after addition of acetone containing 100 μM triticein (final concentration), acetone only, or no addition of solvent. Spores with germ tubes at least as long as half of the spore length were counted as germinated. The data represents spore germination rates in 7 analyzed images per each treatment. Box and whisker plots denote (from top to bottom) maximum, upper quartile, median, lower quartile, minimum. Two-tailed *t*-test *p*-val = 7.86 × 10⁻⁶. **c** Distribution of germ

tube lengths of germinated *B. cinerea* spores, 5 h after addition of acetone containing 100 μM triticein (final concentration), acetone only, or no addition of solvent. *n* = number of individual germinated spores analyzed. Box and whisker plots denote (from top to bottom) maximum, upper quartile, median, lower quartile, minimum. Two-tailed *t*-test *p*-val = 3.23 × 10⁻¹³. **d** Bacterial growth inhibition assays of methicillin-sensitive *Staphylococcus aureus* (MSSA) and *E. coli* strain NR698, in liquid LB media containing 50–200 μM triticein (top panels), or 1–25 μM triticein (bottom panels). Mean values ± SEM are shown. Each time point respectively represents three or four biological replicates in the 1–25 μM, or 50–200 μM triticein experiments. Source data are provided as a Source Data file.

Compared with tetraploid wheat, the hexaploid *T. aestivum* is known to have broader climate adaptability and enhanced tolerance to various biotic and abiotic stresses[37,38]. Here we show that the emmer-*Aegilops* hybridization introduced an isoflavone biosynthetic pathway into the ancestral tetraploid wheat, exemplifying how hybridization-derived polyploidy can lead to enrichment of specialized metabolism in a given plant species through the introduction of an entire metabolic pathway (Fig. 6a). Introduction of the capacity to produce triticein, and possibly other phytoalexins from acquiring the D subgenome, may also have contributed to enhanced tolerance to pathogens in hexaploid wheat.

In silico analyses of publicly available transcriptomic data have previously shown that the BGC4(5D) cluster genes are induced by infection with fungal pathogens[22,39]. Here, qRT-PCR and metabolic analyses of *Fusarium culmorum*-infected wheat plants confirmed that the cluster is pathogen-responsive, supporting a role for triticein as a phytoalexin. Further metabolic analysis of MeJA-treated wheat plants showed that triticein can be detected in various wheat tissues, with highest accumulation in blades (Supplementary Fig. S25). The antifungal and antibacterial activity exhibited by triticein in vitro, and its higher abundance in blades compared to other tissues, suggests a possible role in protection from bacterial and/or fungal foliar pathogens. However, wheat mutants of the triticein pathway did not show increased susceptibility to the fungal pathogens *F. culmorum, B. graminis*, or *M. oryzae* in disease assays. Additional research is thus needed to establish whether the triticein pathway contributes to wheat resistance against other fungal or bacterial pathogens.

Isoflavonoid metabolism has been best characterized in the legumes, in which the step catalyzed by isoflavone synthase defines a key branching point from the flavonoid pathway[40]. Biosynthesis of isoflavones such as daidzein and genistein in soy involves two steps: the 2-hydroxylation and aryl migration of the flavanone substrates to give a 2-hydroxyisoflavanone, catalyzed by the CYP93C enzyme 2-hyroxyisoflavanone synthase (also known as isoflavone synthase; IFS)[18–20], followed by dehydration of the flavanone by 2-hydroxyisoflavanone dehydratase (HID), a carboxylesterase-like protein[41]. In the wheat triticein pathway, however, a different isoflavone biosynthetic route occurs, in which an initial flavanone 2′-hydroxylation reaction is catalyzed by a CYP71 enzyme, followed by aryl migration and C-ring desaturation by the isoflavone synthase (also a CYP71) to give the isoflavone product (Fig. 6b). While there are differences between the two routes, some commonalities are notable: the chemical logic behind the construction of the isoflavone core in the triticein pathway and in legume isoflavones is the same, and is obtained through the rearrangement of a flavone precursor via aryl migration. Also, in both routes the aryl migration is preceded by, or coupled with a cytochrome P450-catalyzed hydroxylation. Also notable is the difference in distribution of the soy and wheat isoflavone synthases; while the soy IFS homologs are widely distributed in the legumes, and are even conserved in the taxonomically distant sugar beet[19], the specialized triticein-related isoflavone synthase is seemingly restricted to species within the *Triticum/Aegilops* D lineage. An extensive phylogenetic analysis of CYP71 sequences from various Poaceae species did not identify apparent orthologs of either

*TaCYP71FS3* or *TaCYP71C164* in other grasses (Supplementary Figs. S26, S27).

The wheat isoflavone synthase *TaCYP71FS3* can potentially be utilized in bioproduction of isoflavonoids in a heterologous system such as fermentation of bacteria or yeast, in which leguminous IFS expression was shown to constitute a pathway bottleneck[42]. Similarly, additional enzymes from the triticein cluster may prove to be useful for engineered production of intermediates or side-products of the triticein pathway, several of which have been investigated for pharmacologically related or other bioactivities, including isosakuranetin (**2**)[43], naringenin 4′,7-dimethyl ether (**3**)[44] and naringenin trimethyl ether (**7**)[45]. It should be noted that the phytoestrogen-related health promoting properties associated with some isoflavones (e.g. genistein, daidzein) do not necessarily apply to triticein, and characterizing potential health-related bioactivities will require further investigation of this compound. Nevertheless, discovery of an isoflavone synthase in wheat paves the way to biofortification of this important food crop with isoflavonoids, through the implementation of genome editing, cisgenics or intragenics, while avoiding necessary introduction of transgenes. Indeed, our analysis of cv. Chinese Spring plants revealed that triticein already naturally occurs in dry grains. Subsequent analyses showed that triticein is also found in grains of several commercial hexaploid wheat cultivars, as well as in commercial wheat flours, albeit only in low quantities (up to 0.1 μg·g⁻¹, in wheat bran) (Supplementary Fig. S28). In addition to the low quantities to which triticein and other naturally occurring isoflavones accumulate in wheat grains, our current lack of knowledge about the regulation of triticein biosynthesis (and about wheat isoflavone metabolism in general) poses a considerable challenge for obtaining high isoflavone-producing grains via GM techniques. The natural occurrence and wide variability in triticein abundance in grains that we observed in only a small set of analyzed cultivars (>40-fold difference between the highest and lowest producer), suggest that isoflavone content in wheat grains is a trait that may also be modified by traditional breeding. This variability also provides an interesting avenue for future research with regard to the regulatory mechanisms underlying triticein accumulation. Interestingly, in MALDI mass spectrometry imaging (MALDI-MSI) of wheat grains cross sections (cv. Avalon), triticein was found solely in the outer bran layer (Fig. 5c, Supplementary Fig. S29), a localization that coincides with its potential protective role against microbial pathogens. Other bran-bound bioactive phenolics are similarly considered to have protective roles in cereal grains[46,47].

Although dozens of biosynthetic gene clusters producing various terpenes, alkaloids, and other classes of plant specialized metabolites have now been characterized[48], the wheat triticein BGC, to the best of our knowledge, is the first functionally validated plant BGC encoding production of a flavonoid. The lack of known flavonoid-producing BGCs is particularly intriguing considering the wide structural diversity and ubiquity of flavonoids in the plant kingdom. Given the wealth of plant phenylpropanoid BGCs predicted in silico[49], and the lack of any apparent reason for why clustering of genes involved in flavonoid biosynthesis would be exceptional, we envision that additional such clusters will be discovered in the future.

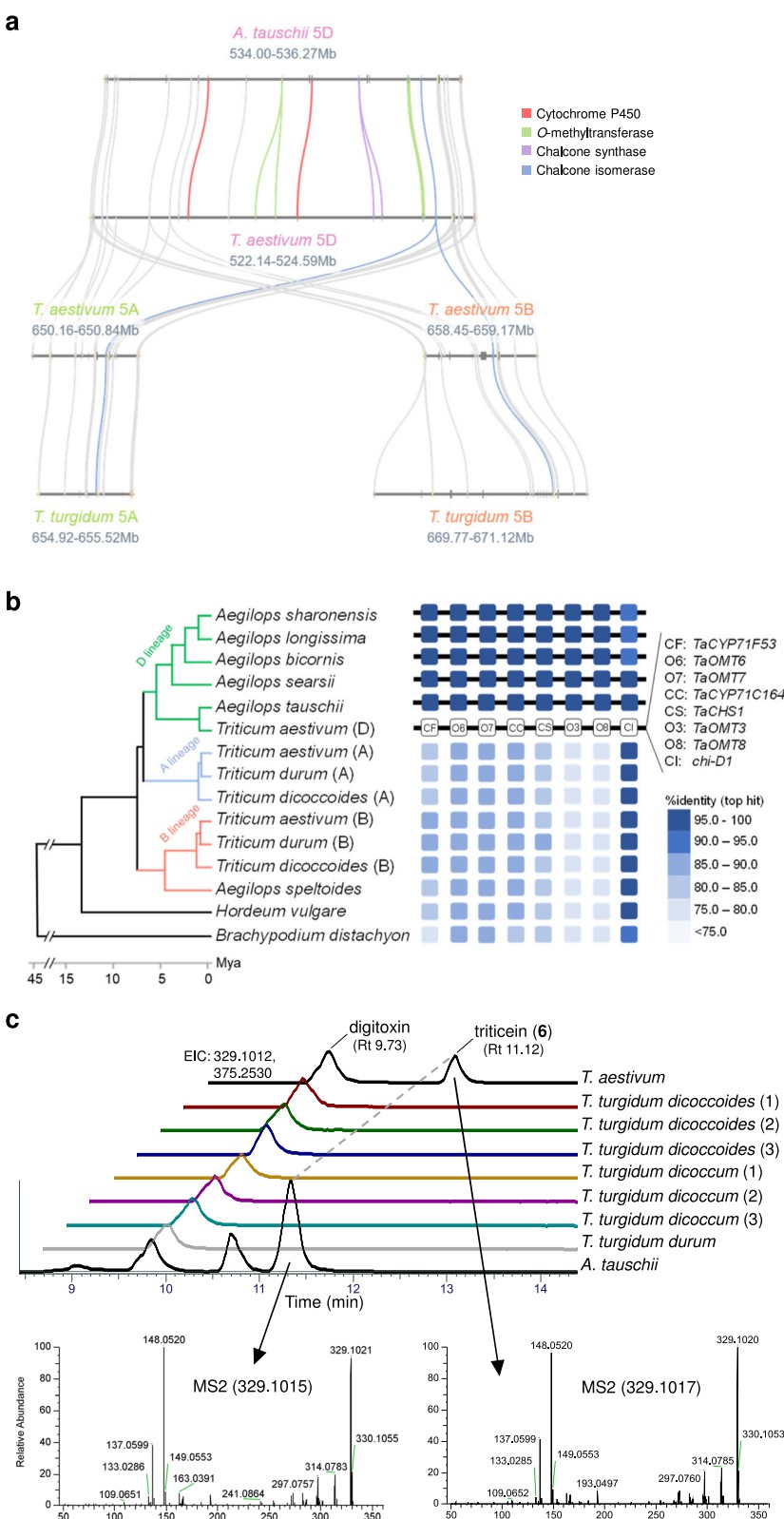

**Fig. 5 | Occurrence of triticein gene cluster in wheat and other cereal genomes.** **a** Microsynteny analysis of triticein BGC in wheat group 5 chromosomes, and syntenic regions in *Aegilops tauschii* and wild emmer wheat (*T. turgidum*). **b** The triticein BGC is conserved in *Aegilops tauschii* and four *Aegilops* grasses from section Sitopsis, but absent in A and B lineage genomes, including wild emmer wheat (*T. turgidum ssp. dicoccoides*), durum wheat (*T. turgidum ssp. durum*), and putative B genome progenitor, *Aegilops speltoides*. Color coding shows % identity of nucleotide sequence of top blastn alignment with the coding sequences of the respective BGC4(5D) genes. The phylogenetic tree is adapted from Li et al.[36]. **c** Extracted ion chromatogram (EIC) and MS2 fragmentation for triticein (*m/z* = 329.1012) in bread wheat, emmer wheat and *Aegilops tauschii*. Digitoxin (*m/z* = 375.2530) was used as an internal standard. *Y*-axes of all chromatograms are linked. Cultivars/accessions analyzed, in order shown in figure from top to bottom: *T. aestivum* Cadenza; *T. turgidum* TRI 18539, PI 428127, PI 466991, CGN 16073, CGN21064, TRI 6141, Miradoux; *A. tauschii subsp. strangulata* TOWWC0190.

## Methods

### Materials and germplasm

Analytical standards were purchased from Biosynth (sakuranetin, iso-sakuranetin, naringenin 4',7-dimethyl ether, and naringenin trimethyl ether) and Sigma (naringenin). Seeds of Chinese Spring, Avalon, Miradoux, Santiago, Soissons, Cadenza, Paragon, and Robigus wheat cultivars were obtained from the John Innes Centre Germplasm Resource Unit (GRU). Seeds of *T. turgidum* accessions TRI 18539, PI 428127, PI 466991, CGN 16073, CGN21064, TRI 6141 and *A. tauschii subsp. strangulata* accession TOWWC0190 were obtained from Cristobal Uauy's group, Department of Crop Genetics, John Innes Centre. Commercial wheat products were purchase from stores in Norwich, UK: plain and wholemeal flour 1 (ASDA brand), plain and wholemeal flour 2 (Allinson's), bran 1 (Neal's Yard), bran 2 (Whole Food Earth), germ 1 (Tree of Life), germ 2 (Holland & Barrett).

### Generation of DNA constructs

For cloning of wheat genes, RNA was extracted from leaves of 10-day-old *Triticum aestivum* plants (cv. Chinese Spring), infected with powdery mildew (*Blumeria graminis f. sp. tritici*)[22], using RNeasy plant mini kit (Qiagen). RNA was treated with RQ1 DNAse (Promega) and cDNA libraries prepared with Superscript IV or Superscript III reverse transcriptase kits (Thermo Fisher Scientific), using oligo(dT) primers, according to manufacturer's protocols. *TaOMT3* (GenBank accession ON108662), *TaOMT6* (ON108663), *TaOMT8* (ON108664), *TaCYP71C164* (ON108660), and *TaCYP71F53* (ON108661) were amplified from cDNA using Phusion DNA polymerase (Thermo Fisher Scientific) or Q5 DNA polymerase (New England Biolabs). *TaCHS1* (ON108659) and *chi-D1* (JN039039) were synthesized by Twist Bioscience, San Francisco, CA, USA. *TaOMT7* was derived from *TaOMT6* coding sequence by site directed mutagenesis[50] to obtain a single mutation (G72R). *AtMYB12* (O22264) was cloned from *Arabidopsis thaliana* col-0 rosette leaf cDNA. Synthesized and cDNA-amplified genes were cloned into a pDONR207 Gateway entry vector and subcloned into a pEAQ-HT-DEST1 plasmid[51], using BP and LR Clonase II enzyme mixes (Thermo Fisher Scientific), respectively. Oligonucleotides used for amplification and sub-cloning are specified in Supplementary Table S3.

### Transient expression in *N. benthamiana* and metabolite extraction

Transient heterologous gene expression in *N. benthamiana* generally followed a previously reported method[52,53]. pEAQ-HT-DEST1 binary expression vectors were transformed into *Agrobacterium tumefaciens* GV3101 via electroporation. Agrobacteria cultures were grown overnight in 28 °C in LB media and resuspended in MMA buffer (10 mM $MgCl_2$, 10 mM MES pH 5.6, 200 μM acetosyringone) to $OD_{600}$ 0.2 and incubated at room temperature for 2–3 h. For co-expression of multiple genes, $OD_{600}$ 0.2 cultures of strains carrying expression vectors for the different genes were mixed 1:1 prior to infiltration. Cultures were infiltrated by needleless syringe into leaves of 5 weeks old greenhouse-grown *N. benthamiana* plants. Control leaves were infiltrated with agrobacteria harboring an empty pEAQ-HT-DEST1 vector. The plants were further maintained in the greenhouse after infiltration. Infiltrated leaves were harvested 5 days post infiltration, freeze-dried and ground. For LC-MS analysis, 250 mg samples were extracted with 4 mL methanol at room temperature for 1 h. Extracts were fully evaporated in a Genevac EZ-2 Elite centrifugal evaporator (SP), resuspended in 200 μL methanol, and filtered through a mini column (pore size 0.22 μm, Geneflow).

### Large-scale agroinfiltration, extraction, purification, and NMR analyses

Vacuum-mediated large-scale agroinfiltrations of *N. benthamiana* plants were based on a previously described method[53,54]. Specific methods for extraction, purification, and NMR analyses of the purified compounds are detailed in Supplementary Data 1.

### Wheat infection with *Fusarium culmorum*

*Fusarium culmorum* (isolate FC2021) was grown in Petri dishes with media comprised of 18 g·L[1] bactoagar in water containing 20% V8 vegetable juice, in a 16 h/ 8 h light/dark photoperiod at 22 °C. 'Chinese Spring' wheat plants were grown in peat-based soil in a controlled environment (16 h/ 8 h light/dark photoperiod, 22 °C). ~60 3-week-old plants were infected with *F. culmorum* inoculum, following a similar previously described method[55,56]. The inoculum was prepared by macerating the *F. culmorum*-containing agar into a thick slurry that was applied onto the plants. A mock infection was carried out on the same number of plants, with a slurry made of agar media without *F. culmorum*. The *F. culmorum* and mock slurries were held in contact with the plants using 3.5 cm length pieces of plastic tubing that were previously placed around the seedlings stem base, into which the slurry was inserted using a syringe. Treated plants were kept in a controlled environment (16 h/ 8 h light/dark photoperiod, 22 °C) in covered trays. Blade samples were collected for RNA extraction and subsequent qRT-PCR analysis 3- and 7-days post inoculation (dpi), and for metabolite analysis 7- and 14-dpi. For both RNA and metabolite extractions, four biological replicates were collected for each timepoint, each replicate containing blade tissues from five plants.

### Metabolite extraction from *F. culmorum*-infected wheat plants

200 mg freeze-dried samples were extracted in 4 mL ethyl acetate containing 80 μg·mL$^{-1}$ digitoxin (Sigma Aldrich) as internal standard in 25 °C for 1 h. Following removal of cell debris by centrifugation, the extract was evaporated in a Genevac EZ-2 Elite centrifugal evaporator (SP), re-dissolved in 300 μL methanol, and filtered through a mini column (pore size 0.22 μm, Geneflow).

### Quantitative real-time PCR (qRT-PCR) of *F. culmorum*-infected wheat plants

For qRT-PCR analysis of wheat plants inoculated with *Fusarium culmorum*, RNA was extracted from 20 mg frozen tissue with RNeasy kit (Qiagen), with on-column DNAse treatment using RNase-free DNAse set (Qiagen), according to manufacturer's protocol. 1 μg of RNA from each sample was reverse-transcribed with High Capacity cDNA Reverse Transcription kit (Applied Biosystems) using oligo(dT) primers, in a 20 μL reaction, according to manufacturer's protocol. Each cDNA was diluted 1:3 in water prior to use in qRT-PCR. All oligonucleotides (Supplementary Table S3) were designed using Primer3 software[57]. qRT-PCR was performed on a CFX96 Touch Real-Time PCR instrument (Bio-Rad) in the following conditions: initial step in the thermal cycler for 3 min at 95 °C, followed by PCR amplification for 40 cycles of 10 s at 95 °C and 30 s at 59 °C, and finally dissociation analysis to confirm the specificity of PCR products with 0.5 °C ramping from 55 °C to 95 °C. Each 10 μL reaction was comprised of 5 μL LightCycler 480 SYBR Green I Master mix (Roche Life Science), 1 μL cDNA template, 3 μL $H_2O$ and 1 μL primer mix (0.5 μM each primer). Relative transcript levels were calculated according to the Pfaffl method[58], using the housekeeping gene β-tubulin (TUBB) as reference[59].

### Cytochrome P450 expression in yeast and in vitro enzymatic assays

*TaCYP71C164* and *TaCYP71F53* were respectively cloned into pAG423-His and pAG425-Leu yeast expression vectors, from a pDONR207 entry plasmid via Gateway cloning (Invitrogen). The expression vectors were transformed into *Saccharomyces cerevisiae* strain WAT11 following the Pompon et al.[60] protocol, and microsomal fractions were isolated based on a previously described method[61], with modifications. Briefly, yeast cells were grown in 10 mL SGI media at 30 °C for 24 h, followed by 1:50 dilution into 250 mL YPGE media and growth in 30 °C for 36 h.

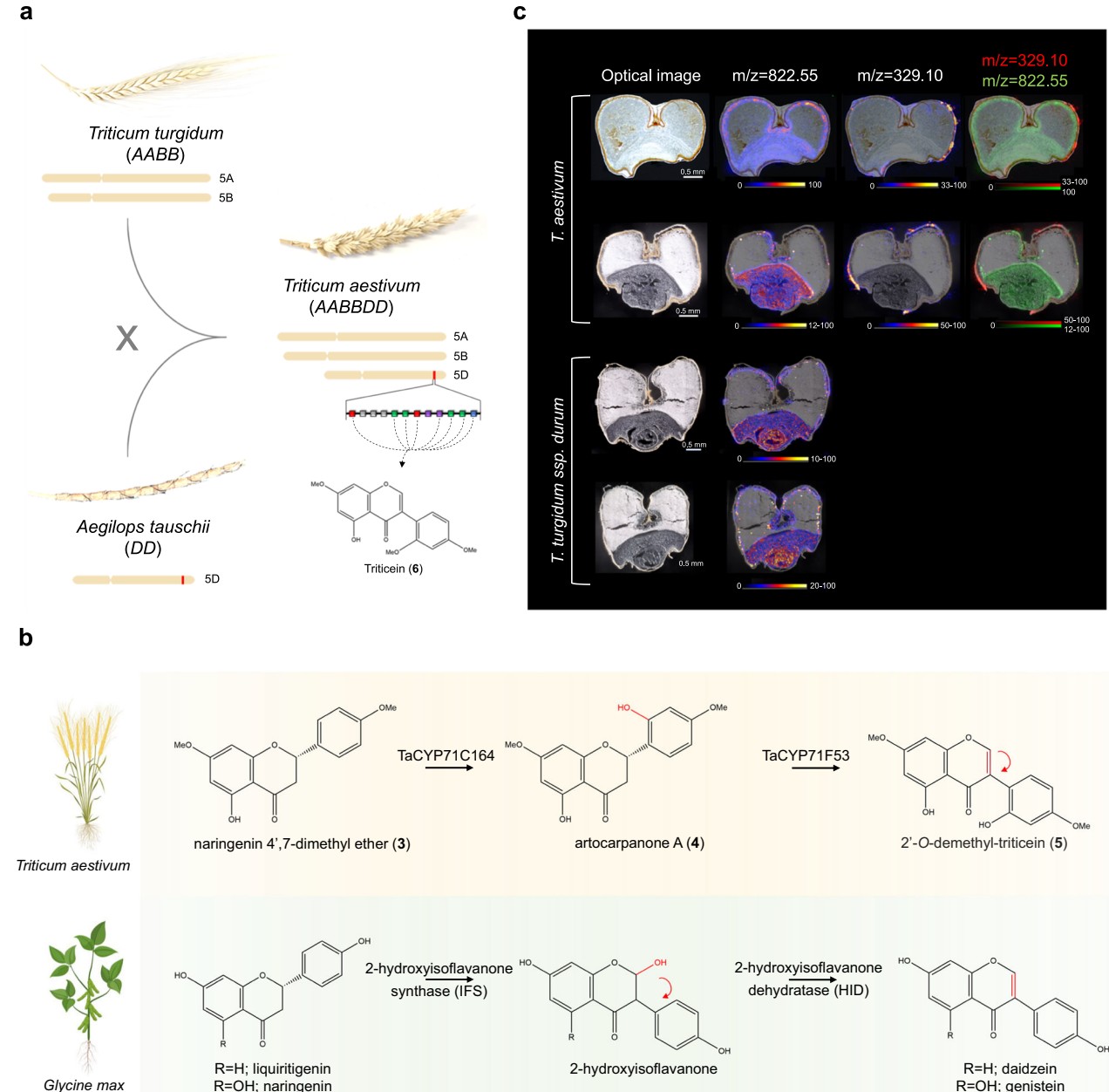

**Fig. 6 | An isoflavone biosynthetic pathway was introduced into bread wheat through a natural hybridization event. a** hybridization with the D genome donor *Aegilops tauschii* resulted in acquisition of a genomic locus encoding an isoflavone biosynthetic pathway into *Triticum turgidum* (emmer), the domesticated progenitor of modern-day bread wheat. **b** biosynthesis of wheat triticein isoflavone differs from biosynthesis of isoflavones produced in soy and other legumes. In wheat, the isoflavone 2'-*O*-demethyl-triticein (**5**) is formed by successive reactions of two cytochrome P450s of the CYP71 family: first 2'-hydroxylation of naringenin 4',7-dimethyl ether (**3**) by TaCYP71C164, followed by aryl migration coupled with C-ring desaturation by TaCYP71F53. In soy, production of daidzein or genistein is respectively obtained by aryl migration and 2-hydroxylation of liquiritigenin or

naringenin, by a CYP93C cytochrome P450, 2-hydroxyisoflavanone synthase (IFS), and subsequent dehydration of the flavanone by 2-hydroxyisoflavanone dehydratase (HID), a carboxylesterase-like protein. Created with Biorender.com. **c** MALDI mass spectrometry imaging (MSI) shows the triticein mass signal (*m/z* = 329.10) is distributed in the outer bran layers of cross-sectioned *T. aestivum* cv. Avalon grains (marked with white arrows) and is undetected in *T. turgidum subsp. durum* (cv. Miradoux) grains. An *m/z* = 822.55 ion highly abundant in germ and aleurone/nucellar layers is shown for reference, putatively identified as a phosphatidylcholine phospholipid, based on literature[75,76]. Color scales indicate intensity of the extracted ion, normalized to total ion current. Colors on high end of scales represent all pixels with n to 100 percent of maximal measured intensity.

Induction was achieved by the addition of a sterile 10% (v/v) solution of 200 g·L⁻¹ galactose and growth for an additional 16 h. Harvested cells were resuspended in 10 mL TEK buffer (0.1 M KC1 in TE) and left at room temperature for 5 min, centrifuged and resuspended in 3 mL TES B buffer (0.6 M sorbitol in TE). The cells were next lysed by hand shaking for 5 min after the addition of glass beads (0.45 mm diameter, Sigma-Aldrich) and centrifuged at 4000 rpm for 10 min, 4 °C,

and the supernatant then used for isolation of microsomes via ultra-centrifugation for 1 h at 30,000 rpm at 4 °C. The microsomal fractions were finally resuspended in 200 µL TE buffer and directly used for in vitro enzymatic assays. 100 µL enzymatic reactions were incubated at 28 °C for 1 h, each reaction containing: 20 µL 0.1 M phosphate buffer pH 7.4 (20 mM final concentration), 10 µL NADPH (Sigma Aldrich) (10 mM final concentration), 10 µL substrate stock solution (1 mM

naringenin 4',7-dimethyl ether (**3**), or 1 mM artocarpanone A (**4**)), 25 μL microsome fraction, and 35 μL double distilled water. Reactions were terminated by the addition of 100 μL methanol. The samples were next evaporated in a Genevac EZ-2 Elite centrifugal evaporator (SP), resuspended in 100 μL methanol and filtered through a mini column (pore size 0.22 μm, Geneflow) prior to LC-MS analysis.

## KASP genotyping and metabolite analysis of wheat TILLING lines

For Kompetitive Allele-Specific PCR (KASP) genotyping, genomic DNA was extracted from blade tissues sampled from wheat seedlings (lines Cad0227, Cad1793, Cad1682, and Cad1684 of a Cadenza TILLING population), using an extraction based on a previously reported protocol[62]. KASP genotyping was carried out as described in the KASP user guide (LGC), using a PACE mix (3CR Bioscience) and oligonucleotides specified in Supplementary Table S3. Genotyping showed that the mutant allele-of-interest is fixed in the Cad0227 line, but not in the remaining three lines, which included plants homozygous or heterozygous for the respective mutant allele, or homozygous for the wildtype allele. For the metabolite analysis, genotyped plants found to be homozygous for the mutant allele were grown in peat-based soil in a greenhouse, together with plants from the Cadenza parental line. Five-week-old plants were sprayed with a 200 μM methyl jasmonate + 0.01% Tween-20 solution. Blade tissues were sampled four days after MeJA treatment, in four to six biological replicates, each containing blades from one plant (Cad1793, Cad1682, and Cad1684) or two plants (Cad0227, Cadenza parent). For metabolite analysis, 150 mg samples of freeze-dried material were extracted with 5 mL ethyl acetate containing 20 μg·mL$^{-1}$ digitoxin (internal standard) at 28 °C for 1 h. After centrifugation and removal of cell debris, 3.5 mL supernatant was evaporated in a Genevac EZ-2 Elite centrifugal evaporator (SP), resuspended in 350 μL methanol, and filtered through a mini column (pore size 0.22 μm, Geneflow).

## *Fusarium culmorum* and *Magnaporthe oryzae* susceptibility assays of TILLING lines

Blade sections (6–8 cm) were sampled from the second leaf of 11-day old, soil-grown seedlings, derived from KASP-genotyped parents that were homozygous for the wildtype or mutant allele. The sections were placed in 10 × 10 cm square clear plastic plates containing 50 mL 1% water agar, in five replicate plates, each containing one leaf section from each genotype. Two rectangular strips (2 × 10 cm) were cut from the centre of the agar and placed on top of the edges of the leaf sections, to form an air gap below the central part of the sections. For the Fusarium assay, *F. culmorum* (isolate FC2021) was grown in Petri dishes with media comprised of 18 g·L$^{-1}$ bactoagar in water containing 20% V8 vegetable juice, in a 16 h/ 8 h light/dark photoperiod at 22 °C. The Fusarium conidia were washed from two-week-old plates using water and a glass rod, filtered through muslin gauze and diluted in water to a working concentration of 1 × 10$^6$ conidia per mL. Two 10 μL drops of the *F. culmorum* suspension were placed on the adaxial side of each section, on top of punctures made by a glass pipette. The cereal blast assay was based on a previously described method[63]. *M. oryzae* (isolate BTJ4P) was grown on 1.5% CMA in 22 °C, 16/8 h light/dark. The conidia from two-week-old plates were washed and diluted to a concentration of 1.5 × 10$^5$ conidia per mL water. Leaves were spray inoculated using a Clarke Wiz Mini Air Compressor spray gun kit (Clarke Tools, Dunstable, England). The lids of the plates were misted with sterile dH$_2$O to increase humidity, and then laid flat in plastic trays lined with blue roll drenched in water to create a humid environment. Trays were placed into an autoclave bag and kept in the dark for 24 h after inoculation, and then incubated at 22 °C under a 16/8 h light/dark photoperiod. Disease phenotype was scored five days post infection, based on a scale of 0 (lowest susceptibility) to 6 (highest susceptibility)[64].

## Powdery mildew susceptibility assays of TILLING lines

Detached leaves from 11-day-old soil-grown Cadenza wheat seedlings were infected with *Blumeria graminis f. sp. tritici* (isolate CAW14S6313, maintained on the susceptible wheat cv. Cerco). Blade segments of ~3 cm length were placed in boxes containing water with 0.5% agar and 100 mg L$^{-1}$ benzimidazole, and were inoculated by blowing fresh spores into settling towers placed over the plant material, according to a previously reported method[65]. Following inoculation, plant material was kept in a growth cabinet at constant temperature of 15 °C and 16 h day-length. Leaves were photographed and colonies counted 7 days post infection.

## Metabolite analysis of MeJA-treated detached wheat leaves

Three cm blade sections were cut from the first leaf of 10-day-old 'Chinese Spring' seedlings grown in peat-based soil. Sections were kept in H$_2$O in a Petri dish for 24 h in a 22 °C lighted growth cabinet (16 h/8 h light/dark photoperiod), then transferred to Petri dishes containing 150 μM methyl jasmonate (Sigma-Aldrich W341002) with 0.02% Tween-20, or H$_2$O with 0.02% Tween-20, and kept in same conditions for three days. Four biological replicates, each comprised of ~25 MeJA- or H2O-treated blade sections were subsequently used for metabolite extraction and LC-MS analysis. 100 mg aliquots of freeze-dried samples were extracted in 4 mL methanol containing 10 μg·mL$^{-1}$ digitoxin (Sigma Aldrich) for 2 h at 25 °C. Following removal of cell debris, the extracts were evaporated in a Genevac EZ-2 Elite centrifugal evaporator (SP), redissolved in 100 μL MeOH, and filtered through a mini column (pore size 0.22 μm, Geneflow), prior to LC-MS analysis.

## Metabolite analysis of MeJA-treated whole wheat plants

For blade, sheath, and root samples, 'Chinese Spring' plants were soil-grown in 11 cm diameter pots in an air-conditioned greenhouse. Five-week-old plants were sprayed with water containing 200 μM methyl jasmonate (Sigma Aldrich W341002) and 0.01% Tween-20. Samples were collected in three biological replicates, each containing tissues from at least six plants. For spikes and rachis samples, post-anthesis spikes from 17-week-old greenhouse-grown 'Chinese Spring' plants were placed in 15 mL conical tubes with water, in a 22 °C lighted growth cabinet (16 h/8 h light/dark photoperiod). The spikes were sprayed with 200 μM methyl jasmonate + 0.01% Tween-20 solution, and rachis or whole-spike samples were collected after four days, in three replicates, each containing four spikes. For seedling samples, 'Chinese Spring' seeds were germinated on wet filter paper in a Petri dish. One day after germination, 2 mL of 200 μM methyl jasmonate + 0.01% Tween-20 solution was added to the plates, which were sealed with parafilm and placed in a 22 °C lighted growth cabinet (16 h/ 8 h light/dark photoperiod). Samples were collected four days after treatment in three biological replicates, each containing 12 whole seedlings. For grain samples, three biological replicates, each containing 0.7 g of dry 'Chinese Spring' grains were ground and used for metabolite extraction and analysis. All collected wheat tissues were lyophilized before extraction. 200 mg dry weight samples were extracted with 8 mL methanol containing 40 μg·mL$^{-1}$ digitoxin for 1 h at room temperature. Cell debris was removed by centrifugation and supernatant was evaporated in a Genevac EZ-2 Elite centrifugal evaporator (SP), resuspended in 200 μL methanol, and filtered through a mini column (pore size 0.22 μm, Geneflow).

## Metabolite analysis of wheat grains and flour

Three technical replicates of 500 mg (plain and wholemeal flours, ground wheat grains) or 100 mg (bran and germ) were extracted in 4 mL methanol containing 20 μg·mL$^{-1}$ digitoxin as internal standard for 2 h with continuous shaking at 37 °C. After centrifugation, 3.5 mL supernatant was evaporated in a Genevac EZ-2 Elite centrifugal evaporator (SP), resuspended in 400 μL methanol, and filtered through a mini column (pore size 0.22 μm, Geneflow).

### LC-MS analyses of flavonoids

High-resolution mass spectrometry analysis of the metabolites was carried out on a Q Exactive instrument (Thermo Scientific). Chromatography was performed using a Kinetex 2.6 μm XB-C18 100 Å, 50 mm × 2.1 mm (Phenomenex) column kept at 30 °C. Water containing 0.1% formic acid (FA) and acetonitrile containing 0.1% formic acid (FA) were used as mobile phases A and B, respectively with a flow rate of 0.4 mL/min. A gradient elution program was applied as follows: 0–0.75 min linearly increased from 0 to 10% B, 0.75–13 min linearly increased from 10 to 60% B, 13–13.25 min linearly increased from 60 to 80% B, 13.25–14.25 min linearly increased from 80 to 100% B, 14.25–14.5 min linearly decreased from 100 to 10% B hold for 2.5 min for re-equilibration, giving a total run time of 17 min. MS detection was performed following electrospray ionization (ESI) in both positive and negative modes, in the range of 100–1500 $m/z$. Photodiode-Array (PDA) detection was recorded in a 200–600 nm range using a vanquish detector (Thermo Scientific).

### Triticein relative and absolute quantification

Quantitative and semi-quantitative LC-MS analyses of triticein (**6**) in wheat extracts (various tissues of MeJA-treated plants, blades of *F. culmorum*-infected plants, grains of commercial cultivars, commercial wheat flours), were carried out using the Xcalibur software package v4.3 (Thermo Scientific). Automatic peak detection and integration was done with ICIS or Genesis algorithms, applying 7 smoothing points and other default parameters, using [M + H]⁺ ion $m/z = 329.1013$ for triticein, and ions $m/z = 375.2530$, $635.3776$, or $747.4305$ for the internal standard, digitoxin (URL: https://massbank.eu/MassBank/RecordDisplay?id=MSBNK-Waters-WA000567). Relative quantification of triticein (**6**) and naringenin (**1**) in agroinfiltrated *N. benthamiana* was carried out in the same method, using ICIS peak detection with 1 smoothing point and default parameters, and the respective ions 329.1021 [M + H]⁺, 787.4243 [M+Na]⁺, 273.0763 [M + H]⁺ and 435.1288 [M + H]⁺ for triticein, digitoxin (internal standard), naringenin, and naringenin-glycoside. For absolute quantification of triticein in commercial wheat flours, a standard curve was generated; a stock solution of 0.1 mg·mL⁻¹ triticein in methanol was prepared, and a series of ten 1:4 dilutions in methanol were prepared from this stock solution. The triticein solutions were next evaporated and extracted with 3.5 mL methanol containing 20 μg·mL⁻¹ digitoxin. The extracts were evaporated in a Genevac EZ-2 Elite centrifugal evaporator (SP), and redissolved in 400 μL methanol, used in LC-MS analysis.

### Spore germination assay

*Botrytis cinerea* strain B05.10 was grown on PDA media (potato dextrose agar) at 22 °C. Conidia collected from plates were diluted in PDB media (potato dextrose broth), to a concentration of $1 \times 10^6$ conidia per 1 mL. 1 μL of a 10 mM triticein solution in acetone was added to 99 μL of the conidial suspension, to give a final concentration of 1% acetone, 100 μM triticein. Control experiments included 1% acetone in conidial suspension, or a suspension with PDB media only. The solutions were placed in capped sterile Eppendorf tubes and incubated without shaking for 3–6 h at 23 °C. 10 μL samples from the solutions were placed on microscope slides, and images (792.45 μm × 792.45 μm) were taken with an Axio Zoom.V16 microscope (Zeiss). Germ tube length measurements were done with Digimizer (MedCalc Software).

### Bacterial growth inhibition assays

Analyzed bacterial strains were streaked out to single colonies on LB agar plates and grown overnight at 37 °C. Colonies were used to inoculate fresh liquid LB media (10 mL) and grown overnight at 37 °C with 250 rpm shaking. Cultures grown overnight were sub-cultured (1:100) into 10 mL fresh LB and grown at 37 °C and shaking to an $OD_{600}$ of 0.6. Cultures were then diluted to $OD_{600}$ 0.001 for the experiment. For the growth inhibition assays, 200 μL incubations were prepared in

96-well plates (Sterilin microtiter U well plates, Thermo Scientific). Each well contained 196 μL bacterial suspension in LB media with initial $OD_{600}$ of 0.001, and 4 μL from triticein stock solutions in DMSO. Control wells contained 4 μL DMSO without triticein. 75 μL mineral oil (Aldrich) was added to each well to prevent evaporation. Each plate contained three or four replicate wells for the 1–25 μM or 50–200 μM triticein experiments, respectively. The bacteria were grown at 37 °C for 12 h in a FLUOstar Omega Microplate Reader (BMG Labtech), under constant shaking at 500 rpm, with $OD_{600}$ readings taken every 30 min.

### Cryosectioning and MALDI mass spectrometry imaging

Cross sections of dry wheat grains (cv. Avalon and Miradoux) were scanned with matrix-assisted laser desorption ionization mass spectrometry imaging (MALDI-MSI). The grains were soaked in water overnight before sectioning. The hydrated grains were frozen in dry ice and embedded onto a flat metal holder with M1 embedding matrix (Thermo Scientific). The grains were sectioned with a Cryostar NX70 Microtome (Thermo Scientific) to give 20 μm-thick sections at −20 °C, which were mounted onto SuperFrost Plus adhesion slides (Thermo Scientific) and dried for 10 min at room temperature.

Optical images were taken using a Canon 5D Mark IV camera with a Canon MP-E 65 mm f/2.8 1–5x Macro Photo lens. Sections were covered with 2,5-dihydroxybenzoic acid matrix (DHB) to a density of ~3 μg·mm⁻² using a SunCollect MALDI Sprayer (SunChrome) with a DHB solution of 10 mg·mL⁻¹ in 80% methanol/0.05% trifluoroacetic acid (TFA). MALDI imaging was performed with a Synapt G2-Si mass spectrometer with a MALDI source (Waters) equipped with a 2.5 kHz Nd:YAG (neodymium-doped yttrium aluminum garnet) laser operated at 355 nm. Mass calibration was performed using clusters of red phosphorous in the range $m/z$ 50–1200. The slides were fixed in the instrument metal holder and were scanned with a flat-bed scanner (Canon). The images were used to generate pattern files and acquisition methods in the HDImaging software version 1.4 (Waters) with the following parameters: area of a complete section ~20 mm², laser beam diameter at low setting (60 μm) with 30 or 45 μm step size, resulting in ~16,000 pixels per section, MALDI−MS-positive sensitivity mode, $m/z$ 50–1200, scan time 0.5 s, lazer repetition rate 1 kHz, lazer energy 200. Total scan time for a complete section was ~4 h. The MS raw files were processed in HDI1.4 with the following parameters: detection of the 1,000 most-abundant peaks, $m/z$ window 0.05, MS resolution 10,000. The processed data were loaded into HDI1.4 and normalized by total ion content (TIC). Images were generated using the HotMetal2 color scale and exported as png image files. The color scales were adjusted as shown in the images for best visualization.

### Blastn search for BGC4(5D) homologs in grasses

Wheat BGC4(5D) homologs in selected grasses were searched with Blastn on the GrainGenes Blast service (https://wheat.pw.usda.gov/blast/), using gene coding sequences as query. The following genome assemblies were used for the blast search: *Triticum turgidum ssp. dicoccoides* WEWseq v2.0[66], *Triticum turgidum ssp. durum* Svevo rel. 1.0[67], *Aegilops longissima*[68], *A. longissima, A. bicornis, A. speltoides, A. sharonensis, A. searsii*[36], *A. tauschii* Aet5.0[69], *Triticum aestivum* IWGSC Refseq v2.1[70], *Hordeum vulgare* Morex v3[71]. For the *Brachypodium distachyon* v3.1 assembly[72], a blastn search was carried out on the JGI Phytozome13 database (https://phytozome-next.jgi.doe.gov/blast-search).

### Microsynteny analysis

To perform microsynteny analysis and generate figures, a python implementation of MCScan[73], https://github.com/tanghaibao/jcvi/wiki/MCscan-(Python-version), was used. FASTA and GFF3 files were retrieved from EnsemblPlants (http://plants.ensembl.org) for chromosomes 5A, 5B and 5D of *Triticum aestivum* (IWGSC), 5A and 5D of *Triticum turgidum subsp. diccocoides* (WEWSeq_v.1.0) and 5D of

*Aegilops tauschii* (Aet_v4.0). MCScan ortholog finding and synteny assignment was run with a c-score of 0.99 and a single iteration.

### Gene expression analysis of *TaCHS1* and *chi-D1* homologs

Putative chalcone-flavanone isomerases (CHI) in wheat were identified by searching on Interpro (URL: https://www.ebi.ac.uk/interpro/) for all proteins containing Pfam domain PF02431, and mapping them to the IWGSC refseq v1.1 assembly. Putative chalcone synthases (CHS) were selected based on a recently reported genome-wide analysis of wheat type-III polyketide synthases[74]. The 10 *CHI* and 25 *CHS* identified genes were assigned to gene expression modules of a previously generated weighted gene co-expression analysis (WGCNA) dataset[22].

### Reporting summary

Further information on research design is available in the Nature Portfolio Reporting Summary linked to this article.

### Data availability

All data supporting the findings of this study are available within the paper and its Supplementary Information. The coding sequences of the genes investigated in the study have been uploaded to NCBI GenBank, the accession numbers are: *TaOMT3* (ON108662); *TaOMT6* (ON108663); *TaOMT8* (ON108664); *TaCYP71C164* (ON108660); *TaCYP71F53* (ON108661); *TaCHS1* (ON108659); *chi-D1* (JN039039). Source data are provided with this paper.

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

## Acknowledgements

We thank Lionel Hill and Paul Brett from the JIC metabolomics platform for assistance with metabolite analyses, JIC horticultural services staff for assistance with plant cultivation, Noam Chayut, Liz Sayers and Simon

Orford (JIC Germplasm Resource Unit) for providing seeds of wheat cultivars, Cristobal Uauy and Nikolai Adamski for helpful discussions and providing seeds of emmer wheat and *Aegilops tauschii*, Jie Li and Eva Wegel for advice on microtome sectioning, Richard Goram and Sophie Preston for assistance with KASP genotyping. G.P. was supported by a Royal Society Kohn International Fellowship (NIF\R1\180677) and a Marie Skłodowska-Curie Individual Fellowship (838242). A.O.'s and B.W.'s labs are supported by the Biotechnology and Biological Sciences Research Council (BBSRC)-funded Institute Strategic Programme (ISP) Grant 'Molecules from Nature' (BB/P012523/1), the BBSRC ISP Grant 'Harnessing Biosynthesis for Sustainable Food and Health (HBio) (BB/X01097X/1), and the John Innes Foundation. P.N.'s group is supported by the BBSRC Cross-Institute Strategic Programme Grant 'Designing Future Wheat' (BBS/E/J/000PR9780).

## Author contributions

G.P. and A.O. conceived the project. G.P. and R.C.M. generated DNA constructs, designed, performed, and analyzed experimental results of heterologous gene expression in *N. benthamiana* and yeast, in vitro enzymatic assays, metabolic analyses by LC-MS/PDA, quantitative real-time PCR, large-scale infiltrations, and wheat MeJA treatments. A.E.D. purified compounds from infiltrated *N. benthamiana* and performed NMR analyses. C.O. and G.P. performed bioinformatic analyses. G.P., L.C., and A.S. performed Fusarium/blast/powdery mildew infections and pathogen susceptibility assays of wheat TILLING lines. H.P.M., R.C.M., and G.P. designed and performed bacterial growth inhibition assays. J.W. performed Botrytis spore germination inhibition experiments, generated microscopy images, and analyzed the data. G.S. and C.M. performed mass spectrometry imaging of grain sections that R.C.M. prepared. B.W., P.N., and A.O. supervised research. G.P. and A.O. wrote the manuscript, with input from all authors.

## Competing interests

A.O., G.P., R.C.M., and A.E.D. are inventors on a patent application arising from this work. The remaining authors declare no competing interests.
