## [Peer Review File · Nature Communications]

Discovery of Isoflavone Phytoalexins in Wheat Reveals an Alternative Route to Isoflavonoid BiosynthesisEditorial Note: This manuscript has been previously reviewed at another journal that is not operating a transparent peer review scheme. This document only contains reviewer comments and rebuttal letters for versions considered at *Nature Communications*.

REVIEWERS' COMMENTS

Reviewer #1 (Remarks to the Author):

The revised manuscript by Polturak et al. responded the comments of the three reviewers and performed additional analysis of the key concerns and provides additional supporting information. The quality of the revision was significantly improved and most of my concerns were correctly addressed.

I have only one further comment for the current manuscript. The authors carried out powdery mildew infection experiments and observed no difference between the mutants and control lines, same as the results of Fusarium and Blast assays. Since no significant disease resistance difference related to the BGC4, I strongly suggest the authors to tone down the claim of the defense-related role of Tritice in wheat.

Reviewer #2 (Remarks to the Author):

I would start my review by stating that I have previously reviewed the manuscript prior to transfer. I feel that Polturak et al. have done an excellent job of responding to the earlier concerns I raised. I also feel that the work is very well executed and written up and that it provides considerable novel insight into isoflavone phtoalexins in wheat. I have no further suggestions for improvement of this manuscript.

Response to reviewer comments- Polturak et al., *Nature Communications*

(Revision for manuscript NCOMMS-23-32240-T)

Referees' comments:

Referee #1 (Remarks to the Author):

The revised manuscript by Polturak et al. responded the comments of the three reviewers and performed additional analysis of the key concerns and provides additional supporting information. The quality of the revision was significantly improved and most of my concerns were correctly addressed.

I have only one further comment for the current manuscript. The authors carried out powdery mildew infection experiments and observed no difference between the mutants and control lines, same as the results of Fusarium and Blast assays. Since no significant disease resistance difference related to the BGC4, I strongly suggest the authors to tone down the claim of the defense-related role of Triticein in wheat.

Authors response:

To address this reviewer request, we have toned down the claim of a defense-related role of triticein in wheat, by editing the text as follows (newly added text in bold):

L42-L43: “Pathogen-induced production and *in vitro* antimicrobial activity of triticein are consistent with **suggest** a defense-related role for this molecule in wheat.”

L429-431: “Additional research is thus needed to ~~understand the potential contribution of~~ **establish whether** the triticein pathway **contributes** to wheat resistance against **other** fungal or bacterial pathogens, ~~and its mechanism of action.~~”

L479-481: “...triticein was found solely in the outer bran layer (Fig. 5C, Supplementary Fig. S29), a localization which coincides with its ~~plausible~~ **potential** protective role against microbial pathogens.”

Referee #2 (Remarks to the Author):

I would start my review by stating that I have previously reviewed the manuscript prior to transfer. I feel that Polturak et al. have done an excellent job of responding to the earlier concerns I raised. I also feel that the work is very well executed and written up and that it provides considerable novel insight into isoflavone phtoalexins in wheat. I have no further suggestions for improvement of this manuscript.